# "It's the poverty"—Stakeholder perspectives on barriers to secondary education in rural Burkina Faso

Jan Jabbarian[1], Luisa Katharina Werner[1,2], Moubassira Kagoné[1,3], Julia Margarete Lemp[1], Shannon McMahon[1,4], Olaf Horstick[1], Harounan Kazianga[5], Jean-François Kobiané[6], Günther Fink[7,8]ᴼ, Jan-Walter De Neve[1]ᴼ*

1 Heidelberg Institute of Global Health, Medical Faculty and University Hospital, University of Heidelberg, Heidelberg, Germany, 2 Faculty of Medicine, Albert-Ludwigs-Universität Freiburg, Freiburg, Germany, 3 Health Research Center of Nouna (Centre de Recherche en Santé de Nouna—CRSN), Ministry of Health, Nouna, Burkina Faso, 4 Bloomberg School of Public Health, Johns Hopkins University, Baltimore, MD, United States of America, 5 Department of Economics and Legal Studies in Business, Spears School of Business, Oklahoma State University, Stillwater, OK, United States of America, 6 Institut Supérieur des Sciences de la Population (ISSP), Université Joseph Ki-Zerbo, Ouagadougou, Burkina Faso, 7 Swiss Tropical and Public Health Institute, Basel, Switzerland, 8 University of Basel, Basel, Switzerland

ᴼ These authors contributed equally to this work.
* janwalter.deneve@uni-heidelberg.de

**Data Availability Statement:** Data requests can be send to the Heidelberg University Hospital Ethics Committee (ethikkommission-I@med.uni-heidelberg.de) by researchers who meet the criteria

## Abstract

Universal primary and secondary education is a key target of the Sustainable Development Goals. While substantial gains have been made at the primary school level, progress towards universal secondary education has slowed, particularly in sub-Saharan Africa. In this study, we aimed to determine perceived barriers of secondary schooling in rural Burkina Faso, where secondary school completion is among the lowest globally (<10%). We conducted a two-stage qualitative study using semi-structured interviews ($N$ = 49). In the first stage, we sampled enrolled students (n = 10), out-of-school adolescents (n = 9), parents of enrolled students (n = 5), parents of out-of-school adolescents (n = 5) and teachers (n = 10) from a random sample of five secondary schools. In a second stage, we interviewed key informants knowledgeable of the school context using snowball sampling (n = 10). Systematic analysis of the pooled sample was based on a reading of interview transcripts and coding of the narratives in NVivo12 using the diathesis-stress model. Recurring themes were classified using *a priori* developed categories of hypothesized barriers to secondary schooling. Major reported barriers included school-related expenses and the lack of school infrastructure and resources. Insufficient and heterogeneous French language skills (the official language of instruction in Burkina Faso) were seen as a major barrier to secondary schooling. Forced marriages, adolescent pregnancies, and the low perceived economic benefits of investing in secondary schooling were reported as key barriers among young women. Our results guide future interventions and policy aimed at achieving universal secondary education and gender equity in the region.

for access to confidential data. Interview transcripts cannot be made publicly available since they contain potentially sensitive information on e.g., out-of-school adolescents.

**Funding:** This work was supported by the Alexander von Humboldt Foundation (https://www.humboldt-foundation.de/), funded by Germany's Federal Ministry of Education and Research (https://www.bmbf.de/en/index.html). JWDN was also supported by the German Research Foundation (405898232) (https://www.dfg.de/en/); NICHD of NIH (R03-HD098982; https://www.nichd.nih.gov/); and the Heidelberg University Excellence Initiative (https://www.uni-heidelberg.de/excellenceinitiative/). The funders had no role in study design, data collection and analysis, decision to publish, or preparation of the manuscript. The contents are the responsibility of the authors and do not necessarily reflect the views of any of the funders or the US government.

**Competing interests:** The authors have declared that no competing interests exist.

# Introduction

Adolescence is a critical period in development when changes in educational processes can have dramatic consequences later in life [1]. Educational investments in adolescence may function as multipliers for future investments [2], suggesting that higher human capital at younger ages increases future acquisition of human capital [3]. During the period of adolescence, a secondary school degree can be compared to a "make-or-break achievement", in which adolescents sort into tracks–such as a career vs. being a stay-at-home parent–leading to divergence and clustering of multiple outcomes subsequently in life [4, 5]. Secondary schooling is a major determinant of long-run health and economic outcomes, including childbearing [6], HIV infection risk [7], labor market participation [8], as well as offspring mortality [9]. Investments in secondary schooling can transform the lives of adolescents, their families, and generate high societal returns [10]. In recognition of this critical period, the United Nations' *Global Strategy for Women's, Children's and Adolescent's Health* (2016–2030) placed adolescence at the heart of the Sustainable Development Goals (SDGs) [11].

Secondary school completion, however, has remained persistently low in many settings in sub-Saharan Africa (SSA) [12], with women attaining lower average education across central and western SSA compared to men. Hypothesized barriers to secondary school include liquidity constraints [13], distance to school [14], lack of information on the benefits of education [15–17], lack of qualified teachers and learning support [18], malnutrition [16], work-related factors (such as caring for sick family members) [17], early pregnancy, as well as other socio-cultural factors [19]. These large existing gaps and inequalities in secondary school completion are likely to be further exacerbated by the COVID-19 pandemic and worsening security situation, which have resulted in losses of income and temporary school closures in many settings, including in Burkina Faso [20, 21]. Relatively little remains known, however, regarding barriers to secondary schooling, particularly in contexts where schooling is very low. For example, in a recent review of interventions to remove barriers to girls' schooling in low- and middle-income countries only two out of >80 studies were conducted in Burkina Faso [22]. Both studies assessed broad efforts to improve the school environment, such as constructing entire schools and additional classrooms, and were focused on the primary school level [23, 24].

In this qualitative study, we conducted semi-structured interviews with a wide range of stakeholders to elicit the perceived barriers to secondary school in rural Burkina Faso, where secondary school completion rates are among the lowest globally (<10%) [25]. We defined stakeholders broadly as those who "can effect or [are] affected by the achievement of the organization's objectives" [26]. We included students, out-of-school adolescents, parents of students, parents of out-of-school adolescents, teachers, as well as other relevant stakeholders. In contrast to prior studies from low- and middle-income countries [27, 28], we included adolescents and youth who are *out-of-school* and their parents. In doing so, we included the perceptions of some of the most disadvantaged households [29]. We conducted field work for the study in a Health and Demographic and Surveillance System (HDSS) area in rural Burkina Faso [30]. The HDSS site is ideally positioned to support our study question because it places adolescents within the context of their families and the broader community. Our overarching aim was to gain a better understanding of the perceived barriers to secondary schooling completion and inform future intervention studies to assess the most effective strategies to increase secondary school enrolment in the study area [31, 32].

# Materials and methods

## Study area

The Nouna HDSS area is located in the Kossi province in the north-west of Burkina Faso (Boucle du Mouhoun region), about 300 km from the capital Ouagadougou. The HDSS site

has existed since 1992 and currently covers a population of $>$105,000 habitants living in 11,750 households [19]. The Nouna HDSS area spans 1,775 $km^2$ and includes the semi-urban village of Nouna (29% of the population) as well as 58 villages (71% of the population) with about 30 secondary schools (see reference [30] for a map). The mostly rural population consists predominantly of subsistence farmers and cattle keepers. The Nouna HDSS site is operated by the Nouna Health Research Centre (CRSN), funded by the Burkinabe Ministry of Health [30]. The main ethnic groups are the Bwaba, Dafing, Mossi, Peulh, and Samo. While the official language of instruction in the study area is French, the Dioula language serves as a 'lingua franca', permitting communication between the different ethnic groups [30]. Additional details on the general population living in the HDSS area are provided elsewhere [16, 30].

## Educational context in Burkina Faso

Formal education in Burkina Faso follows a "6-4-3 system", including 6 years of primary schooling, which grants the "*Certificat d'études primaires*" (CEP), and 4 years of post-primary schooling, which grants the "*Brevet d'études du premier cycle*" (BEPC). In principle, school is mandatory in Burkina Faso for all children ages 6 to 16 years. Senior secondary schooling grants the "*Baccalauréat*" after 3 years and is not mandatory. Burkina Faso had the 8th lowest Education Index globally in 2020 (182nd out of 189 countries) [33]. Gross lower secondary school enrolment was 56% and upper secondary school enrolment was 18% in 2018. Mean years of schooling completed among adults was 1.8 years for men and 1.0 years for women in 2014 [34]. Access to school varies substantially by geographical region [19] and is particularly low in rural areas [25]. The junior secondary school completion rate, for instance, ranges from 4.5% in the rural Sahel region to 43% in the Centre region of Burkina Faso. At the national level, commonly reported reasons for school absenteeism include a lack of financial means and a lack of interest in attending school (school is "not deemed necessary") [35]. These findings suggest that poverty and the perceived benefits of going to secondary school may play an important role in decision-making around school participation and performance [36–38].

## Study design

We conducted a two-stage qualitative study using semi-structured interviews. After a first round of pilot interviews, we improved our study instruments for comprehensibility and, based on preliminary results, further fine-tuned the questionnaires used in the study. In the first stage, we used maximum variation sampling within five randomly sampled secondary schools in the Nouna HDSS area. From each selected school, a total of eight interviewees were selected: two enrolled students, two out-of-school adolescents (who did not attend school between one month and one year prior to the interview), one parent of an enrolled student, one parent of an out-of-school adolescent, as well as two teachers. Respondents were randomly selected by data collectors from the most recent class lists available with the support of school staff as needed (e.g., to reach out-of-school adolescents). In the second stage, we drew from a pool of respondents as informed by guidance from local study team members and complemented this approach with snowball sampling. We developed questionnaires for our qualitative semi-structured interviews separately for each category of stakeholders (e.g., enrolled students, out-of-school adolescents, parents, and teachers). The questionnaires covered hypothesized barriers to secondary schooling, using *a priori* developed categories of barriers (described below), and were based on readings of the literature and prior research in the study area [28–30]. All questionnaires are available in **S1 File**. In addition, we collected basic quantitative socio-demographic characteristics from all participants (such as age and parental education).

## Data collection

**Interviews.** Experienced local interviewers, who were hired and managed directly by the CRSN, were trained in Nouna, Burkina Faso, for two days. Trainings covered the intentions of the study, instruments, the adaptation of interview styles by category of study respondents (e.g., adolescents vs. adults), sensitive topics (e.g., reasons for leaving school early), and the importance of a confidential atmosphere. Face-to-face interviews were held between 14th April and 22nd April 2018 and audio-recorded either in French or a local language according to the interviewee's preference. Written informed consent was obtained from every interviewee or parent for interviewees younger than 18 years. Adult consent was obtained in advance of the child's assent for minors. When showing quotes in the text, we added the first letter of a pseudonym after each quote to protect the privacy of interviewees and allow the reader to track the comments of interviewees across quotes. Interviews conducted in a local language were translated in French and transcribed by local experts affiliated with the CRSN [39, 40].

**Study sample.** We conducted 49 interviews in total. In the first stage, we interviewed enrolled students (n = 10), out-of-school children and youth (n = 9), parents of enrolled students (n = 5), parents of out-of-school children and youth (n = 5) and teachers (n = 10). In the second stage, we interviewed key informants knowledgeable of the school context, including healthcare professionals, caretakers, and parents engaged in local school management (n = 10). Our sample size was guided by the work of Morse 2016: a total of approximately 50 respondents was considered sufficient to gain saturation as access to secondary schooling in Burkina Faso represents mostly explicit, apparent information [41]. One out-of-school adolescent could not be interviewed due to migration out of the HDSS and two transcripts were incomplete.

## Data analysis

Our analysis proceeded in three steps. First, in preparation for our analysis, we reviewed the existing published and grey literature. We conducted multiple rounds of literature searches in PubMed, Google Scholar, SSRN, and ProQuest. The search strategy was conducted iteratively using English and French search terms, beginning with broad search terms (e.g., "secondary school", "barrier", "hindrance", "denial of schooling") and progressively expanded based on findings (e.g., "drop-outs", "stressors", "vulnerability", "diathesis", "Sub Saharan Africa", "Burkina Faso"). We supplemented our search results with several relevant publications through expert consultation [4, 19, 32, 42]. We then reviewed the first 100 titles of articles which included our search terms anywhere in the text. We reviewed full-text versions of all articles whose primary focus was related to the secondary schooling. In total, we reviewed full-text versions of approximately 50 articles, book chapters, and case studies.

Second, we read interview transcripts and coded recurring themes related to barriers to schooling across participants using NVivo12. Building on the diathesis-stress model, our aim was to understand which stressors were reported by respondents [43]. Third, we categorized emerging themes into eight *a priori* developed categories. The main idea was to increase the comparability of our findings to the existing literature on barriers and interventions aimed at improving secondary schooling in the context of poverty. We were interested in barriers not just at school but in the broader environment (e.g., at home or on the way from home to school). Our categories of barriers to schooling were not mutually exclusive so that emerging themes were classified to the closest fit. We decided not to disaggregate our findings by sociodemographic characteristics of respondents (e.g., age) because saturation is unlikely to have been reached in each of these subgroups given the smaller sample size.

## Conceptual framework

Our empirical work builds on the diathesis-stress model, which is more commonly used in the psychiatry and psychology literature [43]. Briefly, the model considers 'stressors' which challenge an individual's resilience defined as the capability to maintain or regain mental stability [44, 45]. Coping of a person can be described as "conscious volitional efforts to regulate emotion, cognition, behavior, physiology, and the environment in response to stressful events or circumstances" [46]. These mechanisms have been categorized in the literature as e.g., adaptive vs. maladaptive strategies. Maladaptive coping strategies have been specified as harming [47]. In our application, we hypothesize that circumstances and individual resilience influence what a barrier to secondary school *means* to households and adolescents, rather than deterministic quantitative criteria. Based on our review of the literature prior to the analysis, we considered a wide range of possible stressors to secondary schooling (displayed in **Fig 1**). Specifically, we considered the following categories of stressors: (i) economic (e.g., school expenses, opportunity costs of attending secondary school), (ii) health (e.g., sexual and reproductive health, sanitation and hygiene), (iii) psychological (e.g., perceived benefits of investing in schooling), (iv) sociocultural (e.g., gender differences), (v) structural (e.g., school infrastructure), (vi) political and legal (e.g., legal age of marriage), (vii) safety and security, as well as (viii) geographic factors (e.g., climate, distance to school). Factors were seen as stressors if they were reported to disrupt schooling immediately and/or may lead to maladaptive coping strategies of households and adolescents, ultimately resulting in early school leaving [48, 49]. For instance, we considered living with a caretaker elsewhere to attend secondary school, due to travel distance to school or security concerns, as a potential stressor which may contribute or lead to leaving school early [50, 51]. We also looked for behaviors we thought could be indicative of maladaptive coping strategies and may cause short term stress relief but have detrimental effects to the individual in the future. Maladaptive coping strategies may include, for instance, substance abuse, risky (sexual) behavior, and social withdrawal.

## Ethical clearance

This study was pre-registered and approved by the Comité Institutionnel d'Ethique du Centre de Recherche en Santé de Nouna (N° 2018-03-/CIE-CRSN) in Burkina Faso, and the Heidelberg University Hospital Ethics Committee (S-193/2018) in Germany.

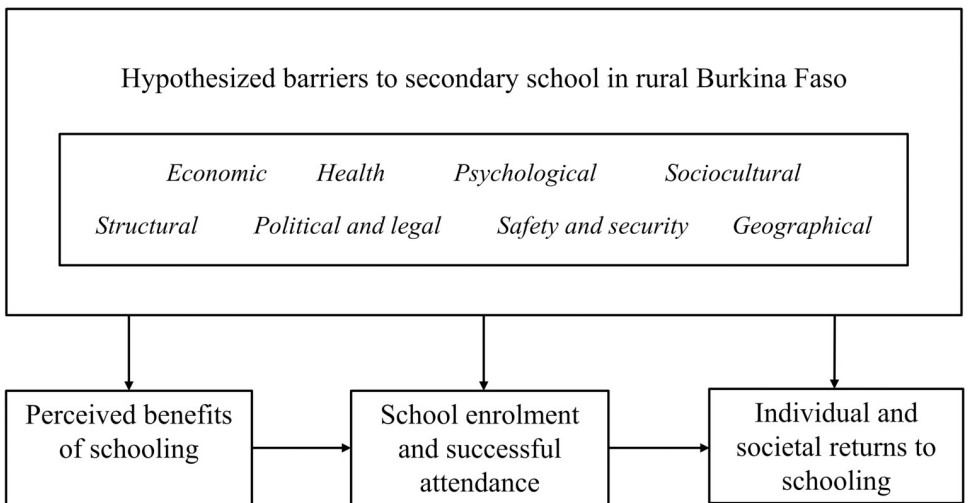

**Fig 1. Conceptual framework underpinning the study.** *Notes*: Fig 1 shows hypothesized barriers or 'stressors' to secondary school in rural Burkina Faso which may affect (i) the perceived benefits of school, (ii) school enrolment and successful attendance, and (iii) the returns to investments in schooling among individuals and their communities.

### Inclusivity in global research

Additional information regarding the ethical, cultural, and scientific considerations specific to inclusivity in global research is included in the **S1 Checklist**.

## Results

### Descriptive statistics

The average age among students and parents in our sample was 17 and 53 years, respectively (**Table 1**). Across all respondents, 26.5% were female, 55.1% were Muslim and 38.8% were

**Table 1. Selected characteristics of study respondents (*N* = 49).**

| *Characteristics* | Children and youth | | Parents | | Teachers and headmasters | Others | Total |
|---|---|---|---|---|---|---|---|
| | Enrolled (n = 11) | Out-of-school (n = 9) | Enrolled (n = 5) | Out-of-school (n = 5) | (n = 13) | (n = 6) | n (%) |
| *Age (n = 49)* | | | | | | | |
| <15 | 2 | 0 | 0 | 0 | 0 | 0 | 2 (4.1) |
| 15–17 | 4 | 6 | 0 | 0 | 0 | 0 | 10 (20.4) |
| 18–20 | 1 | 2 | 0 | 0 | 0 | 0 | 3 (6.1) |
| 21–40 | 2 | 0 | 0 | 1 | 9 | 3 | 15 (30.6) |
| >40 | 0 | 0 | 3 | 3 | 2 | 3 | 11 (22.4) |
| Missing | 2 | 1 | 2 | 1 | 2 | 0 | 8 (16.3) |
| *Sex (n = 49)* | | | | | | | |
| Male | 6 | 5 | 5 | 5 | 10 | 5 | 36 (73.5) |
| Female | 5 | 4 | 0 | 0 | 3 | 1 | 13 (26.5) |
| Missing | 0 | 0 | 0 | 0 | 0 | 0 | 0 (0.0) |
| *Religion (n = 49)* | | | | | | | |
| Christian | 7 | 3 | 0 | 2 | 6 | 1 | 19 (38.8) |
| Muslim | 4 | 6 | 4 | 3 | 6 | 4 | 27 (55.1) |
| Animist | 0 | 0 | 1 | 0 | 1 | 0 | 2 (4.1) |
| Missing | 0 | 0 | 0 | 0 | 0 | 1 | 1 (2.0) |
| *Mother tongue (>1 possible) (n = 49)* | | | | | | | |
| Bwamu | 5 | 3 | 1 | 2 | 2 | 0 | 13 (26.5) |
| Dafing | 4 | 4 | 1 | 1 | 0 | 1 | 11 (22.4) |
| Moore | 0 | 0 | 1 | 1 | 5 | 3 | 10 (20.4) |
| Dioula | 1 | 2 | 1 | 2 | 3 | 1 | 10 (20.4) |
| Others | 3 | 3 | 2 | 1 | 9 | 2 | 20 (40.8) |
| Missing | 1 | 0 | 0 | 0 | 0 | 0 | 1 (2.0) |
| *French (n = 49)* | | | | | | | |
| Yes | 11 | 7 | 0 | 3 | 13 | 4 | 38 (77.6) |
| No | 0 | 2 | 5 | 2 | 0 | 2 | 11 (22.4) |
| Missing | 0 | 0 | 0 | 0 | 0 | 0 | 0 (0.0) |
| *Schooling (years) of own or cared for child (n = 33)* | | | | | | | |
| 1–6 | 3 | 0 | 1 | 1 | 0 | 0 | 5 (15.2) |
| 7–10 | 4 | 6 | 4 | 3 | 0 | 1 | 18 (54.5) |
| 11–13 | 3 | 1 | 0 | 0 | 0 | 1 | 5 (15.2) |
| >13 | 1 | 0 | 0 | 0 | 0 | 0 | 1 (3.0) |
| Missing | 0 | 2 | 0 | 1 | 0 | 1 | 4 (12.1) |

*Notes*: Table 1 shows selected characteristics of study respondents. Data are number of individuals. The category 'others' includes key informants who were interviewed during the second stage of data collection and include adolescents and youth, tutors, teachers, school directors, member of a students' parent association, as well as healthcare professionals.

Christian. The most commonly spoken mother tongues were Bwamu (26.5%) and Dafing (22.4%), in addition to Dioula (20.4%), which 57.1% also spoke as a second language. French was spoken by more than three-fourths (77.6%). Reported school fees per year differed between school level and school type and ranged from 9 to 139 USD per year. Eight out of nine out-of-school adolescents reported being "currently employed" at the time of the study. About two-thirds of the parents interviewed in our sample had no formal schooling themselves, and only one parent had attended secondary school. Maternal occupations included mostly homemaking ("housewife") and farming, whereas fathers were predominantly farmers. In addition to adolescents and youth, parents, and teachers, we also interviewed two school directors, the president of a students' parent association, caretakers with whom students lived to be able to attend a secondary school, and healthcare professionals who worked in the Nouna HDSS area.

## Economic barriers to secondary school

**School-related expenses.** School-related expenses, such as school fees, uniforms, and textbooks, were a recurring theme among all types of stakeholders, including students, out-of-school adolescents and youth, parents, and teachers, and a major perceived barrier to schooling (see **Table 2** for an overview of results). These expenses pushed some household budgets to their limits, were a frequent cause of grade repetition and early school leaving, and posed a major challenge to daily life among adolescents who were able to remain in school. For instance, respondents reported running short of food when school fees were due, and students often did not eat before or during school. As a result, households felt obliged to sell assets (e.g., animals) and adolescents engaged in selling small goods, criminal activities, or sex work to pay for school-related expenses. One young woman reported selling peanuts to pay for school expenses, such as pencils.

*"It's the poverty."*–I, Out-of-school adolescent, female, 17 years

*"(. . .) what really would give me the courage to go to school would be a school free of charge, security, and investments in scholarships. Because today if you see criminal students or students who prostitute themselves this is because not all children have the energy to work during hunger, during hardships. That is why those start committing crimes."*–H, Student, male, 20 years

For parents who were farmers, the annual crop yields and selling prices had a critical impact on their ability to pay for school expenses. Similar situations were reported from small business owners during times of reduced income. Several parents preferred to invest their limited resources in the family's economic activities rather than the secondary schooling of their children. Nevertheless, despite these important challenges to school enrolment and participation, respondents mentioned potential solutions to directly address liquidity constraints. They mentioned lower secondary school fees, merit-based scholarships for students (a system reportedly well established in the past at universities), loans to pay for school expenses, and the payment of school fees in instalments rather than in lump sums.

**Opportunity costs and limited career prospects.** A major source of demotivation was the tension to start work sooner (such as agricultural activities at home) rather than completing secondary school. Students were demotivated by observing out-of-school adolescents who earned a small income and had access to various status symbols (such as a motorcycle). Young men mentioned, for instance, the appeal of planting sesame seeds, working in mines, or

**Table 2. Barriers to secondary schooling reported in rural Burkina Faso.**

| Barrier | Examples |
|---|---|
| *Economic* | • Opportunity costs of schooling (e.g., home production, farming)<br>• Expenditures for school fees and material (e.g., uniforms, books)<br>• Expenditures for transport to school (e.g., bicycle)<br>• Cost of living with a caretaker to attend school<br>• Limited career prospects beyond agriculture<br>• Low salary for teachers |
| *Health* | • General health issues (e.g., malaria infection, headaches)<br>• Early marriage and adolescent pregnancy<br>• Lack of safe drinking water, sanitation, and hygiene<br>• Lack of sex-separated toilets for young women<br>• Food insecurity at home<br>• Malnourishment |
| *Psychological* | • Gender differences in the perceived benefits of education<br>• Temptation to start work sooner<br>• Demotivation due to financial difficulties<br>• Demotivation due to grade repetition<br>• Stigma surrounding pregnancy in school<br>• Preoccupation with lack of childcare options |
| *Sociocultural* | • Schooling of young women seen as "lost investment"<br>• Early and forced marriages<br>• Language of instruction (French) |
| *Structural* | • Home learning environment (e.g., a separate place for learning)<br>• Long distance between home and secondary school<br>• School infrastructure, buildings, classrooms<br>• Standard and status of school building<br>• Supply of drinking water in school<br>• Electricity at home and school (e.g., light, air conditioning)<br>• Teacher qualifications and resources<br>• High student-teacher ratio (>70) |
| *Political and legal* | • Lack of political commitment, enforcement, resources<br>• Education law (e.g., language of instruction) |
| *Safety and security* | • Jihadist threat from national and international terror groups<br>• Lack of security on streets at night |
| *Geographical* | • Travel to/from secondary school<br>• Learning at home and school during the heat<br>• Harmattan winds (hot desert winds carrying dust) |

*Notes*: Table 2 lists barriers or 'stressors' to secondary schooling reported by study participants in rural Burkina Faso (*N* = 49), separately by category of barriers. In S1 Text, we provide additional illustrative quotes.

migrating to Ivory Coast to engage in cacao farming. Adolescents who engaged in paid work sooner appeared to be "rewarded" for leaving secondary school early. In the study area, a large proportion of adolescents is out-of-school which may have further accentuated the temptation among in-school adolescents to start work sooner. Additionally, career prospects were limited by an agriculture-based economy, dominated by subsistence production, and characterized by low crop and livestock productivity. Stakeholders saw few career opportunities other than working in agriculture. These beliefs shaped the view on the relevance of secondary schooling investments among adolescents and youth.

> "*Yes, there is a barrier from something different. (...) The major part of our students has many friends who stay at home do farming and get motorcycles paid. Because they planted sesame, they have motorcycles.*"–E, Teacher, male, 28 years

> "*We don't have other jobs. It's nothing but agriculture here.*"–N, Out-of-school adolescent, male, 17 years

## Health barriers to secondary school

**Sexual and reproductive health concerns.** Teachers reported several health concerns, in particular with regards to sexual and reproductive health. Coming from a poor family put young women at risk to agree on relationships with men, and respondents noted that forced marriages still occurred in the Nouna HDSS area. Early and undesired pregnancies created several barriers to secondary school leading to leaving school early. Two interviewed young women reported to have quit school because of pregnancy. Teachers reported that young women who were pregnant faced teasing and contemptuous language or behavior directed at their pregnancy. Adolescent pregnancy was seen as a mark of disgrace. Similarly, the inability to bring children to secondary school and lack of childcare options for in-school adolescents generated additional worry which distracted young women from active secondary school participation. From a teacher's personal experience, at least one mother was able to continue her schooling while breastfeeding.

> "*As I got pregnant, I said I would drop out this year.*"–J, Out-of-school adolescent, female, 17 years

**Lack of sanitation and hygiene.** Toilets were not present at all school facilities, and students reported that going to the villages to find a toilet was front and center in their minds. At one school, students and staff reportedly used the bushes, which was felt to be profoundly unpleasant by students because of the lack of privacy. Female students also wished for single sex toilets to stop the teasing at toilets between the genders; with the exception of madrassas (Islamic schools), which have guaranteed separate toilets for women and men. Similarly, while running water at school was deemed to be essential, it was not available at every institution. Answers by students and teachers ranged from no running water at school, to wells in the nearby village, to open freshwater barrels. Even if a well was desired, drilling trials did not necessarily turn out to be successful.

> "*At our school here there are no toilets, there is not even a well. (. . .) If you want to drink some water, you have to go to the village. (. . .) if you get diarrhoea you have to go back to the village because here aren't any toilets.*"–E, Teacher, male, 28 years

**Food insecurity and poor nutritional environment.** Food intake among adolescents and youth appeared almost non-existent, disrupted, or highly limited, both at home and at school. At the household level, food shortages due to financial distress among households and adolescents were reported to hinder concentration in school by a wide range of stakeholders, including caretakers, students, and teachers. At the school level, most stakeholders noted that cafeterias were not available at every institution, and having breakfast or lunch depended on the means of parents. Respondents noted that expanding the possibilities of having lunch at or near secondary school (e.g., through newly established school cafeterias or mobile food vendors located close to school) would be a good intervention to reduce school absenteeism by providing a better environment for both learning and recreation and a successful working routine–e.g., school, lunch, and homework. While food availability played an important role, *quality* of diet was not mentioned during the interviews.

> "*What could be brought on way so students can stay in school are school canteens. (. . .) Schools are far away and the parents don't have enough money to give to them so that they can buy something for lunch.*"–O, Parent, male, 51 years

## Social and cultural barriers to secondary school

**Gender discrimination.**   Young women, in particular, were often denied secondary schooling by their parents. One reason appeared to be gender discrimination across several settings, including at home, school, and in the labor market. Young women were encouraged to specialize in domestic skills (homemaking) during early adolescence, which often implied becoming a stay-at-home mom instead of having a career. According to one school headmaster, investments in daughters' schooling paid little dividends in the minds of parents because young women were married to be "housewives" and ultimately leave school early anyway (even if secondary schooling was deemed intrinsically important by parents and young women themselves). In contrast, young men were often allowed to attend secondary school to specialize in skills relevant to the labor market or to be able to take over the family's economic activities in the future.

"*The girls were seen as lost investments. There was no focus on schooling girls. It's true that today the education of girls is encouraged but there are many factors which do not favor the schooling of girls.*"–D, Teacher, female, age unknown

**Language barriers in school.**   Limited and heterogeneous French language skills (the official language of instruction in Burkina Faso) were a key bottleneck to secondary school completion. French language skills among adolescents and youth were repeatedly described by teachers in our sample to be "low", "not tolerable" and, in extremis, as "they know nothing". Teachers mentioned that they need to repeat the same questions again and, in some instances, turned to speaking a more "simplified" French to be understood or switch to local language(s). Schools had implemented a French-only policy on some school grounds to improve student's French skills. However, while students were taught French early in primary school, they would not understand and be able to express themselves well enough in French until at least 9th grade. In addition, the general lack of school libraries was a barrier to mastering French.

"*Right here the real barrier of students is the French language.*"–C, Teacher, female, 26 years

"*One sends a child [to school] who understands nothing but his or her mother tongue. To speak French one has to pass the 1st grade and continue till CEP [graduation from primary school after 6th grade]. It is generally those who succeed in primary school who go further to Collège [junior secondary school]. Those difficulties students have in primary school, have an impact on their time in Collège.*"–P, Parent, male, 54 years

## Psychological barriers to secondary school

**Low perceived benefits of schooling.**   Parents underinvested in the secondary schooling of their children if they perceived few benefits of schooling. While respondents nearly universally reported short-term health benefits to schooling (such as improved sexual and reproductive health), responses regarding economic as well as indirect and long-term health benefits were more ambiguous. Parents reported not sending their children to school based on subjective benefit-cost assessments of attending school and instead instructed them to work at home, in small shops or market stalls, or engage in other paid work. Parents invested money that could have been allocated to school fees in economic or domestic activities (e.g., purchasing animals for farming or breeding). This seemed especially true for farmers' families and for parents without formal schooling themselves. Parents without formal schooling saw limited

benefits of secondary schooling since they had been making their living without having completed formal education themselves.

> "*On Saturdays and Sundays when the children should study for school they have to work on the field.*"–B, Teacher, male, 35 years

### Structural barriers to secondary school

**Poverty and home learning environment.** Not every household was able to provide an environment conducive to learning. Key issues included the lack of a separate workspace and electricity. Students reported that younger siblings challenged their learning when no separate workspace was available at home for studying. Nightfall and battery life were reported by students and teachers as limiting factors to learning since electricity, torches, or solar panels were not widely available at home. Some electrified schools, however, offered students to come and learn for exams at school. Additionally, support with schoolwork could often not be provided at home, possibly because many parents had completed little or no secondary schooling themselves. Instead, relatives or neighbors who were knowledgeable about school topics provided support to students, including homework. The heat in summer further impeded learning at home because air conditioning or ventilators were rarely available similar to most other areas in rural Burkina Faso.

> "*We don't have electricity at home. It is also not a permanent house. It is made of clay. In the last year, I used a flashlight for studying.*"–M, Out-of-school adolescent, male, 17 years

**Long journey to secondary school.** Travel times from home to secondary school ranged from 5 minutes to over 1 hour by foot (see reference [30] for a map of the Nouna HDSS area). A common concern among students and teachers was the frequent breakdowns of bikes and the lack of resources for bike repairs. Left without means of transport, students then either returned home immediately or walked long distances to school. Fatigue from lengthy walks hindered active participation in class.

> "*I wake up at 4 a.m. to help mom with housework. After that I hit the road to school at 6 a.m. I repeat my lessons until the teachers come to class.*"–L, Out-of-school adolescent, female, 17 years

**Lack of school infrastructure and resources.** All categories of stakeholders highlighted the lack of school equipment and material as a barrier to effective school participation and completion. School buildings, for instance, were knocked down by the wind, windows were broken, and roof repairs frequently disrupted class. Classes were sometimes held in a shop at the market or elsewhere outside school to protect students from broken window glass in the classroom. The available furniture at school seemed limited to chairs, benches, and tables of poor quality, and frequently relied on joint teacher-parent projects (co-financed by parents). Students and teachers also complained about lacking access to water, toilets, electricity, and air conditioning or functional ventilators at schools, as noted above. Additionally, schools did not have science labs, which made science lessons only theoretical. Moreover, accessibility to schools for disabled students was highly limited.

> "*The government has adopted a policy of access to education for everyone. But until now we still have classes under palm trees.*"–F, Headmaster, male, age unknown

Teachers deplored the general lack of resources which further limited teaching possibilities and quality. The lack of copiers disrupted the distribution of class materials. Chalk, maps or textbooks were not available in sufficient numbers, suitable storage space for class material was not provided, and computers were not available at all. Higher ranking authorities assessed the school's necessities but shipments with school material would be of low quality, late, insufficient in numbers or incomplete. A teacher gave the example of shipments which provided material for literature classes but omitted materials for mathematics and science. Teachers and students also described classes with up to 100 students per class and sharing a textbook with over 15 students. Overcrowded classes hampered teaching quality and learning in school. The lack of school cafeterias was felt by students, teachers, and parents and further reduced the number of meeting points or workplaces where students could complete their schoolwork.

*"They are with 73 [students]. In that class, students must sit packed. Moreover, this is not easy because of the heat."*—A, Teacher, male, 32 years

## Discussion

Using data from nearly 50 in-depth qualitative interviews with a wide range of stakeholders, we examined perceived barriers to secondary schooling in rural Burkina Faso, where secondary schooling completion rates are among the lowest worldwide [12]. Our study reveals two salient findings. First, respondents reported a wide range of barriers to schooling across sectorial boundaries and layers of society (**Table 2**). Building on the diathesis-stress model, we identified multiple stressors, which either entirely disrupted schooling or challenged the resilience of households and–alone or in combination with other stressors–ultimately led to leaving school early [43]. Second, we find that major perceived barriers included school-related expenses, the lack of school infrastructure and resources, and insufficient and heterogeneous French language skills (the official language of instruction). In addition, forced marriages, adolescent pregnancies, and the low perceived economic benefits of investing in schooling were reported as key barriers among young women. Taken together, many–often unpredictable and uncontrollable–obstacles stood in the way of obtaining a secondary school degree, which diverted substantial cognitive resources on a daily basis. Even with much resilience and excellent coping strategies it remains difficult to overcome such challenges.

While basic education is–in principle–free of charge in Burkina Faso, school-related expenses were a common concern and have been shown to be relevant in many other contexts [52, 53]. Expenses identified in our study included, for example, school fees, uniforms, textbooks, travel and accommodation costs, as well as school furniture. Moreover, many households faced the opportunity costs of sending adolescents and youth to secondary school as opposed to engaging in economic and domestic activities (with financial and time losses in the short term), and the lack of career 'options' beyond engaging in agricultural activities [54]. Respondents brought up potential solutions to address economic barriers to schooling, including school fee reductions, merit-based scholarships, borrowing money, and paying school fees in instalments throughout the year rather than in an annual lump sum. Previous research confirms that school fee reductions may improve human capital and long-term economic growth [55]. Similarly, cash transfer programs have improved several economic and health outcomes [56].

Stakeholders also highlighted the need to improve school infrastructure and resources. Our findings resonate with prior research on the potential positive impact of strengthening "school communities" [57] and positive behavioral changes promoted by schools (e.g., learning

behavior, antisocial and disruptive behaviors [58, 59]). Previous research has also found measurable improvement of reducing class sizes on students' grades in standardized tests, and an increased likelihood to enter tertiary education [60]. However, the benefits of class size reductions may be muted without investments in teaching quality [61], which was relevant in the study context. Quality of school infrastructure and equipment have also been linked with academic outcomes [62, 63]. Rheinländer et al., for instance, highlighted the role of sanitation for girls' secondary schooling rates [64]. A recent review identified several promising interventions to remove barriers to girls' school participation and learning in low- and middle-income countries, including lack of water and sanitation [22]. In terms of nutritional environment, school-based food assistance could provide students to have at least one proper meal a day, further improving both health and education outcomes jointly [23, 65, 66].

A major perceived bottleneck to schooling was the language of instruction. Much of primary education would be dedicated to learning French. The sentiments of teachers in our study were already brought up by Obanya in the 1980s, who noted: "*It has always been felt by African educationists that the African child's major learning problem is linguistic. Instruction is given in a language that is not normally used in his immediate environment (. . .)*" [67]. Brock-Utne called the exclusion of the mother tongue as language of instruction an educational barrier for millions of African children [68]. French language skills varied widely in our study. According to respondents, students would require about six years of schooling to carefully express themselves in French (with little opportunity to do so at home or in school). Previous studies from low-income countries suggest that individuals with fewer than six years of schooling often remain "functionally illiterate and innumerate" [69, 70]. Our findings seem consistent with a report of the Ministry of Education of Burkina Faso, suggesting that just 40.7% of students at the end of CM2 (6[th] grade) score less than 45 out of 100 points in a standardized French test [71]. The recent increase in primary school enrolment in Burkina Faso may have further affected school quality so that students are less well prepared for secondary school [34]. Additionally, our findings relate to barriers to school in other countries in the region, where education at higher levels is generally provided in non-indigenous languages [72].

## Implications for future research

Our findings have several implications for future research. First, there is a need for well-designed studies on tackling barriers to schooling in settings where secondary schooling is persistently low [22, 73]. Our work can inform future schooling interventions which aim to assess strategies to reduce the number of adolescents who leave school early. The 2020 Global Education Evidence Advisory Panel, an interdisciplinary expert panel, reviewed the latest evidence on educational interventions and highlighted several cost-effective strategies to improve schooling outcomes. Providing information on the benefits of education to children and parents, for instance, was identified as a "great buy" [73]. Future interventions could also allow farmers to pay school fees around the time of harvest (when farmers have less liquidity constraints) or reschedule the academic year to lower opportunity costs associated with farming activities. Second, further investigation is needed to distinguish between factors which are "hard" and "soft" barriers to secondary school, where hard barriers may definitively lead to leaving school early (e.g., inability to pay for school expenses) and soft barriers, which either alone or in combination with other factors may lead to leaving school early (e.g., living with a caretaker elsewhere to attend school due to travel distance). To our knowledge, we applied the diathesis-stress model for the first time in the context of barriers to secondary school [43]. Third, climatic factors, such as the Harmattan winds (hot desert winds) were perceived as a barrier to secondary schooling. In the rural Sahel region of Burkina Faso, it is likely that

changes in climate will further worsen schooling outcomes [74]. The relationship between climate shocks and household resilience is an important avenue for future research.

## Study limitations

Besides its strengths, our study has several limitations. First, we randomly selected five out of 28 secondary schools in the Nouna HDSS. The sample we drew was recruited in and around these five study schools, potentially omitting the perceptions of individuals living in underserved villages. Second, the gender distribution was unequal, with 13 female participants vs. 36 male participants. Parents were represented by fathers, possibly due to norms and attitudes which considered the father being head of the family, thereby missing maternal perceptions [75]. Third, the identification of potential study participants in the second stage of the study was guided by local study team members, which may introduce sampling bias into our results. Data collectors may prefer participants who are, for instance, more similar in terms of background characteristics and may be more likely to be available to them. Nevertheless, study participants represented a wide range of demographic characteristics, including with regard to mother tongue and socio-economic status (**Table 1**). Fourth, adolescents and youth may have given socially desired answers or may have been intimidated by the interviewers' seniority. Respondents with formal education were more likely to identify and reflect on barriers to school, while adolescents who were out-of-school required more probing. Fifth, although local professionals translated and transcribed all interviews, the translation of interviews from a local language to French may have led to loss of information. Sixth, since the time of data collection, the security situation has worsened in the study area, and two coups d'état were launched in Burkina Faso (January and September 2022). Our results, particularly around security as a potential barrier to schooling, may not be generalizable to different periods [20].

## Conclusions

Access to secondary schooling in rural Burkina Faso is affected by a wide range of perceived barriers to schooling, including economic, health, psychological, sociocultural, and structural factors. Major perceived factors included the burden of school-related expenses, lack of school infrastructure and resources, and limited and heterogeneous language skills in secondary school. Forced marriages, adolescent pregnancies, and gender differences in the perceived benefits of investing in secondary schooling were key perceived barriers to secondary schooling among young women. Our results guide future schooling interventions and policy aimed at achieving universal secondary education and gender equity in the region.

## Supporting information

**S1 Checklist. Inclusivity in global research.**
(DOCX)

**S1 Text. Additional quotes on barriers to secondary school.**
(DOCX)

**S1 File. Study questionnaires.**
(PDF)

## Acknowledgments

We thank study participants for their time. We also thank Dr. Aurélia Souares and Dr. Lea Jabbarian for helpful comments on earlier drafts of the manuscript. We acknowledge financial

support for the publication fee by the Deutsche Forschungsgemeinschaft within the funding programme "Open Access Publikationskosten" and by the University of Heidelberg.

## Author Contributions

**Conceptualization:** Jan Jabbarian, Luisa Katharina Werner, Günther Fink, Jan-Walter De Neve.

**Data curation:** Jan Jabbarian, Luisa Katharina Werner, Moubassira Kagoné, Günther Fink, Jan-Walter De Neve.

**Formal analysis:** Jan Jabbarian, Luisa Katharina Werner, Julia Margarete Lemp, Shannon McMahon, Olaf Horstick, Harounan Kazianga, Jean-François Kobiané, Günther Fink, Jan-Walter De Neve.

**Funding acquisition:** Jan-Walter De Neve.

**Investigation:** Jan Jabbarian, Luisa Katharina Werner, Shannon McMahon, Olaf Horstick, Harounan Kazianga, Jean-François Kobiané, Günther Fink, Jan-Walter De Neve.

**Methodology:** Jan Jabbarian, Luisa Katharina Werner, Moubassira Kagoné, Julia Margarete Lemp, Shannon McMahon, Olaf Horstick, Harounan Kazianga, Jean-François Kobiané, Günther Fink, Jan-Walter De Neve.

**Project administration:** Moubassira Kagoné, Olaf Horstick, Günther Fink, Jan-Walter De Neve.

**Resources:** Moubassira Kagoné, Günther Fink, Jan-Walter De Neve.

**Software:** Jan Jabbarian, Luisa Katharina Werner, Günther Fink, Jan-Walter De Neve.

**Supervision:** Moubassira Kagoné, Olaf Horstick, Günther Fink, Jan-Walter De Neve.

**Validation:** Jan Jabbarian, Luisa Katharina Werner, Günther Fink, Jan-Walter De Neve.

**Visualization:** Jan Jabbarian, Günther Fink, Jan-Walter De Neve.

**Writing – original draft:** Jan Jabbarian, Günther Fink, Jan-Walter De Neve.

**Writing – review & editing:** Jan Jabbarian, Luisa Katharina Werner, Moubassira Kagoné, Julia Margarete Lemp, Shannon McMahon, Olaf Horstick, Harounan Kazianga, Jean-François Kobiané, Günther Fink, Jan-Walter De Neve.

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
