## [Decision Letter · Decision Letter 0]

3 Jun 2022

PONE-D-21-34581“Girls were seen as lost investments” – stakeholder perspectives on barriers to secondary education in rural Burkina FasoPLOS ONE

Dear Dr. De Neve,

Thank you for submitting your manuscript to PLOS ONE. After careful consideration, we feel that it has merit but does not fully meet PLOS ONE’s publication criteria as it currently stands. Therefore, we invite you to submit a revised version of the manuscript that addresses the points raised during the review process.

The reviewer raised a number of issues that must be addressed, including a need for clarification of some methodological details and reasons for the study design. They also felt that the Introduction and Discussion should be better placed in the context of existing studies. Their comments can be viewed in full below and in the attached file.

We look forward to receiving your revised manuscript.

Kind regards,

Natasha McDonald, PhD

Associate Editor

PLOS ONE

Journal Requirements:

4. We note that Figure S1 in your submission contain map images which may be copyrighted. All PLOS content is published under the Creative Commons Attribution License (CC BY 4.0), which means that the manuscript, images, and Supporting Information files will be freely available online, and any third party is permitted to access, download, copy, distribute, and use these materials in any way, even commercially, with proper attribution. For these reasons, we cannot publish previously copyrighted maps or satellite images created using proprietary data, such as Google software (Google Maps, Street View, and Earth). For more information, see our copyright guidelines: http://journals.plos.org/plosone/s/licenses-and-copyright.

a. You may seek permission from the original copyright holder of Figure S1 to publish the content specifically under the CC BY 4.0 license.  

Reviewers' comments:

Reviewer's Responses to Questions

**Comments to the Author**

1. Is the manuscript technically sound, and do the data support the conclusions?

Reviewer #1: Yes

2. Has the statistical analysis been performed appropriately and rigorously? 

Reviewer #1: N/A

3. Have the authors made all data underlying the findings in their manuscript fully available?

Reviewer #1: Yes

4. Is the manuscript presented in an intelligible fashion and written in standard English?

Reviewer #1: Yes

5. Review Comments to the Author

Reviewer #1: The authors should be commended for putting together this important qualitative work, which will hopefully help inform educational policy in Burkina Faso. Overall this is a good paper, though there are some stylistic and content revisions/additions that I would suggest -- more detail can be found in the Reviewer Attachments. Most/all of the suggestions are ones I believe the authors can address and will make the paper stronger.

6. PLOS authors have the option to publish the peer review history of their article (what does this mean?). If published, this will include your full peer review and any attached files.

Reviewer #1: No

---

## [Author Response · Author response to Decision Letter 0]

28 Jul 2022

Reviewer comments

1.1 Title. The title “Girls were seen as lost investments” is a compelling one, but gives the impression that the paper as a whole will be about barriers to secondary schooling for (primarily) girls. The authors may want to consider using a different quote for their title.

We agree with the Reviewer and have now changed the title as recommended. We believe the new quote in the title better reflects our findings and the paper as a whole:

“It’s the poverty” - stakeholder perspectives on barriers to secondary education in rural Burkina Faso” (p.1, revised manuscript)

1.2 Introduction. Pg 4: Describe upfront who counts as a stakeholder in this context.

We have now clarified the term in the Introduction, as recommended by the Reviewer:

“We defined stakeholders broadly as those who “can effect or [are] affected by the achievement of the organization’s objectives” [23]. We included students, dropout students, parents of enrolled and dropout students, teachers, as well as other relevant stakeholders. In contrast to prior studies in developing settings [24, 25], we included adolescents and youth who are out-of-school and their parents. In doing so, we included the perceptions of some of the most disadvantaged households [26].” (p. 4, revised manuscript)

Reference:

Freeman RE, Reed DL (1983). Stockholders and stakeholders: A new perspective on 

corporate governance. California Management Review, 25(3), 91. 

1.3 Materials and Methods. Pg 5: Under “Study Area”, for the statements about the predominant occupations of residents, ethnic groups, and languages in the Nouna HDSS, are there any citations that you can provide?

We thank the Reviewer for this suggestion and have now provided additional citations. In particular, the article by Sié et al. (2010) provides a detailed HDSS profile. We have added this reference to the revised text, which now says:

“While the official language of instruction in the study area is French, the Dioula language serves as a ‘lingua franca’, permitting communication between the different ethnic groups [27]. Additional details on the general population living in the HDSS area are provided elsewhere [16, 27].” (p. 5, revised manuscript) 

References:

[16] Werner LK, Jabbarian J, Kagoné M, McMahon S, Lemp J, Souares A, et al. "Because at school, you can become somebody" - The perceived health and economic returns on secondary schooling in rural Burkina Faso. PLOS ONE. 2019;14(12):e0226911.

[27] Sié A, Louis V, Gbangou A, Müller O, Niamba L, Stieglbauer G, et al. The Health and Demographic Surveillance System (HDSS) in Nouna, Burkina Faso, 1993-2007. Glob Health Action. 2010;3. doi: 10.3402/gha.v3i0.5284.

Pg 6: Though the authors address in the “Study Limitations” section that there may be bias in their data due to the small sample size, more can be done to address possible sources of bias. For example, providing more detail about the total number of participants invited to interview by participant type (in-school adolescent, out-of-school adolescent, parent, teacher, etc. by gender and other sociodemographic characteristics, if available) and the total number of rejections broken down by those same characteristics would be helpful to understand where bias may lie.

Thank you for this suggestion. We drew the interviewees from five randomly selected secondary schools out of 28 possible secondary schools within the 58 villages in the Nouna HDSS area. Per chosen school 8 interviewees were sampled. They were invited to participate by members of HDSS or school staff. Among those, one selected participant could not be interviewed due to migration of the HDSS area. We have now clarified this in the Methods section of the paper: 

“Respondents were randomly selected by data collectors from the most recent class lists available with the support of school staff as needed (e.g., to reach adolescents and youth who dropped out of school). One dropout could not be interviewed due to migration out of the HDSS and two transcripts were incomplete.” (p.7, revised manuscript)

The sentence in Page 7 under “Study sample” that begins with “Respondents were randomly selected...” would be better placed under the “Study Design” subsection.

We have now moved the sentence as recommended by the Reviewer.

More detail needs to be provided about the second stage sampling – e.g. were there possible sources of bias in the selection of the second stage respondents, given that local study team members were the ones who guided the identification of possible participants?

We have now further clarified the second stage sampling, where we write:

“… the identification of potential study participants in the second stage of the study was guided by local study team members, which may introduce sampling bias into our results. Data collectors may prefer participants who are, for instance, more similar in terms of background characteristics and may be more likely to be available to them. Nevertheless, study participants represented a wide range of demographic characteristics, including with regard to mother tongue and socio-economic status.” (p. 25, revised manuscript)

Describe the pilot study first, prior to any description about the first stage sampling.

We have moved the corresponding sentence as recommended by the Reviewer.

“After a first round of pilot interviews, we improved our study instruments for comprehensibility and, based on preliminary results, further fine-tuned the questionnaires used in the study.” (p. 6, revised manuscript)

Pg 7: – Under “Interviews”, were the local experts hired and managed directly by the study team, or were they associated with a survey firm of some kind?

Local experts were hired and managed directly by the study team as opposed to e.g., a survey firm. We have now further clarified this in the paper, where we write:

“Experienced local interviewers, who were hired and managed directly by the CRSN, were trained in Nouna, Burkina Faso, for two days.” (p. 6-7, revised manuscript)

There is a redundant citation under “Study Sample” ([30]).

Thank you for catching this. We have now removed the redundant citation.

Pg 8: Provide a formal definition of “resilience” and define what “maladaptive coping strategies” are in this context.

We have added definitions for resilience and coping strategies in the Methods section:

“… the model considers ‘stressors’ which challenge an individual’s resilience defined as the capability to maintain or regain mental stability [35, 36]. Coping of a person can be described as “conscious volitional efforts to regulate emotion, cognition, behavior, physiology, and the environment in response to stressful events or circumstances” [37]. These mechanisms have been categorized in the literature as e.g., adaptive vs. maladaptive strategies. Maladaptive coping strategies have been specified as harming [38]. In our application, we hypothesize that circumstances and individual resilience influence what a barrier to secondary school means to households and adolescents, rather than deterministic quantitative criteria.” (p. 8, revised manuscript)

“We also looked for behaviors we thought could be indicative of maladaptive coping strategies and may cause short term stress relieve but have detrimental effects to the individual in the future. Maladaptive coping strategies may include, for instance, substance abuse, risky (sexual) behavior, and social withdrawal.” (p. 9, revised manuscript)

1.4 Result. General comments: First, it’s unclear where many of the accounts from - students, out of school adolescents, parents, teachers, or others. This is important for some of the views on barriers to education (e.g. the accounts of students being envious of out-of-school peers in the labor force). If the descriptions came from multiple types of stakeholders, it may benefit the reader to be more explicit.

Apologies for the lack of clarity. We have now clarified the type of stakeholder throughout the Results section – e.g.:

“Teachers reported school absenteeism due to several health concerns, in particular with regards to sexual and reproductive health.” (p. 13, revised manuscript)

“Female students also wished for single sex toilets to stop the “teasing” at toilets between the genders; with the exception of madrassas (Islamic schools), which would have guaranteed separate toilets for women and men.” (p. 14, revised manuscript)

“Nightfall and battery life were reported by students and teachers as limiting factors to learning since electricity, torches, or solar panels were not widely available at home.” (p. 18, revised manuscript) 

Second, a clearer distinction should be made about whether some barriers primarily affect boys or affect boys and girls more-or-less equally. For example, the account about the salience of the benefits of not being a student by observing the relative wealth of out-of-school peers seems like a barrier that affects mostly boys, since girls are expected to be homemakers according to the interviews. The language used when describing this observation is gender-neutral, even though the barrier is not.

We thank the Reviewer for this suggestion and have now clarified this in the paper – e.g.:

“Young men mentioned, for instance, the appeal of planting sesame seeds, working in mines, or migrating to Ivory Coast to engage in cacao farming.” (p. 13, revised manuscript)

“Early and undesired pregnancies created several barriers to secondary school leading to early school dropout of girls. Two interviewed young women reported to have quit school because of pregnancy. Teachers reported that young women who were pregnant faced teasing and contemptuous language or behavior directed at their pregnancy.” (p. 13, revised manuscript)

Lastly, there is an excessive use of the word “would” in much of this section. Many of the accounts are of events or observations that have already happened, but this passive language makes it sound like some of these events have not happened yet. The language needs to be amended.

We have now reduced the use of the word “would” in this section – e.g.:

“Households and adolescents saw few career opportunities other than working in agriculture.” (p.13, revised document)

“Young women were encouraged to specialize in domestic skills (homemaking) during early adolescence, which often implied becoming a stay-at-home mom instead of having a career.” (p.16, revised document)

“According to one school headmaster, investments in daughters’ schooling paid little dividends in the minds of parents because young women were married to be “housewives” and ultimately drop out of school anyway …” (p.16, revised document)

Pg 12: There is a typo: “–;” should be changed to “–”.

We have now corrected the typo in the paper.

Pg 13: – Does the statement “Early and undesired pregnancies created several barriers to secondary school leading to early school dropout of girls,” refer only to girls in the sample or female students more generally?

We have now clarified this point in the paper. We obtained confirmation of two young women in our sample to have quit school as a direct result of becoming pregnant:

“Early and undesired pregnancies created several barriers to secondary school leading to early school dropout of girls. Two interviewed young women reported to have quit school because of pregnancy.“ (p. 14, revised manuscript)

The quote used in this section is not directly related to sexual or reproductive health outcomes. A quote from a female adolescent (either a current student or an out-of school student) about pregnancy would be a better fit for this section.

We thank the Reviewer for this excellent suggestion. We have now used an alternative quote which more directly illustrates the gist of this sub-section in the paper: 

“As I got pregnant, I said I would drop out this year.” – Dropout, female, 17 years

Pg 14: There is a typo: “non-existing” should be changed to “non-existent”.

We thank the Reviewer for bringing this to our attention. This word has been corrected.

Pg 15: “...labor market skills” to “...skills relevant to the labor market”.

The sentence has changed as suggested.

Pg 16: “...hypothetical school fees” to “...money that could have been allocated to school

fees”.

The sentence has now been edited accordingly.

When referring to “the family business”, is it the case that every family in the sample had their own enterprise?

We thank the Reviewer for this comment. The reported kind of work included subsistence farming, or work in small shops or small market stalls e.g. to sell food. Whether this in every case was owned by the family or family members were employed is not clearly distinguishable in every case. We have now clarified this in the paper, where we write:

“Parents reported not sending their children to school based on subjective benefit-cost assessments of attending school and instead instructed them to work at home, in small shops or small market stalls, or engage in other paid work. Parents invested money that could have been allocated to school fees in economic or domestic activities (e.g., purchasing animals for farming or breeding).” (p. 17, revised manuscript)

Pg 17: More justification is needed for how “Motivational constraints” is different than “Op-

portunity costs,” given that this section is very clearly describing the opportunity costs of being in school for some students. Personally, I would remove this section and place some of the content into the “Opportunity costs” section.

We agree with the Reviewer and have moved the text to the section on opportunity costs. 

Pg 18: Change “...held in a shop at the market or outside school to e.g. protect students” to

“...held in a shop at the market or elsewhere outside school to protect students”.

The sentence has now been edited accordingly.

1.5 Discussion. General comments: This may help with the background section as well, but there are many quasi-experimental or experimental studies that have attempted to address the barriers to schooling that have not been included as references in this paper. Psaki et al. (2022) “Policies and interventions to remove gender-related barriers to girls’ school participation and learning in low- and middle-income countries: A systematic review of the evidence” (link) has a fairly comprehensive list of these types of studies. Though not all are secondary school-specific, hopefully it will be useful.

We thank the Reviewer for pointing us to this helpful reference. We now cite it and discuss several papers from this review in our Discussion section of the paper – e.g.:

“Rheinländer et al., for instance, highlighted the role of sanitation for girls’ secondary schooling rates [55]. A recent systematic review identified several promising interventions to remove barriers to girls' school participation and learning in low-resource settings, including lack of water and sanitation [56].” (p. 23, revised manuscript)

“… there is a need for well-designed studies on tackling barriers to schooling in settings where secondary schooling is persistently low [56, 66].” (p. 24, revised manuscript)

“In terms of nutritional environment, school-based food assistance could provide students to have at least one proper meal a day, further improving both health and education outcomes jointly [57-59].” (p. 24, revised manuscript)

References:

Aurino E, Gelli A, Adamba C, Osei‐Akoto I, Alderman H (2020). Food for thought? Experimental evidence on the learning impacts of a large‐scale school feeding program in Ghana. Journal of Human Resources. doi: 10.3368/jhr.58.3.1019-10515R1.

Kazianga H, de Walque D, Alderman H (2009). Educational and health impacts of two school feeding schemes: Evidence from a randomized trial in rural Burkina Faso. World Bank, 2009. doi: 10.1596/1813-9450-4976.

Psaki S, Haberland N, Mensch B, Woyczynski L, Chuang E (2022). Policies and interventions to remove gender‐related barriers to girls' school participation and learning in low‐ and middle‐income countries: A systematic review of the evidence. Campbell Systematic Reviews; 18(1). doi: 10.1002/cl2.1207

---

## [Decision Letter · Decision Letter 1]

6 Sep 2022

PONE-D-21-34581R1“It’s the poverty” – stakeholder perspectives on barriers to secondary education in rural Burkina FasoPLOS ONE

Dear Dr. De Neve,

Thank you for submitting your manuscript to PLOS ONE. After careful consideration, we feel that it has merit but does not fully meet PLOS ONE’s publication criteria as it currently stands. Therefore, we invite you to submit a revised version of the manuscript that addresses the points raised during the review process.

We look forward to receiving your revised manuscript.

Kind regards,

Miquel Vall-llosera Camps

Senior Editor

PLOS ONE

Journal Requirements:

Reviewers' comments:

Reviewer's Responses to Questions

**Comments to the Author**

1. If the authors have adequately addressed your comments raised in a previous round of review and you feel that this manuscript is now acceptable for publication, you may indicate that here to bypass the “Comments to the Author” section, enter your conflict of interest statement in the “Confidential to Editor” section, and submit your "Accept" recommendation.

Reviewer #1: All comments have been addressed

Reviewer #2: (No Response)

Reviewer #3: (No Response)

2. Is the manuscript technically sound, and do the data support the conclusions?

Reviewer #1: Yes

Reviewer #2: (No Response)

Reviewer #3: Yes

3. Has the statistical analysis been performed appropriately and rigorously? 

Reviewer #1: N/A

Reviewer #2: Yes

Reviewer #3: N/A

4. Have the authors made all data underlying the findings in their manuscript fully available?

Reviewer #1: Yes

Reviewer #2: Yes

Reviewer #3: Yes

5. Is the manuscript presented in an intelligible fashion and written in standard English?

Reviewer #1: Yes

Reviewer #2: Yes

Reviewer #3: Yes

6. Review Comments to the Author

Reviewer #1: Thank you all for addressing my comments. I think this is a nice improvement on the previous version of the paper and you should be commended for your hard work.

Reviewer #2: Girls were seen as lost investments” – stakeholder perspectives on barriers to secondary education in rural Burkina Faso

Thank you for giving me the opportunity to review this article. It is an interesting article and relevant for not only the context of Burkina Faso but also for similar countries. The paper is conceptually sound and well-written, but I think it needs some minor revisions before it can be published. The following are my suggestions.

“These large existing gaps and inequalities in secondary school completion are likely to be further exacerbated by the COVID-19 pandemic, which has resulted in dramatic losses in the income and livelihoods of poor households [19] and temporary school closures in many settings [20].”. This point is relevant, but COVID 19 impact seems to be relatively less a major issue in most African countries, including Burkina Faso. I would suggest also adding and discussing an aspect which is more relevant to education, that is insecurity in many areas of the country. Currently, security issues seem to be more relevant to education than COVID 19 is. How does that affect girls’ education?

After reading the introduction, I could not find out the rationale for the study. In relation to the topic, what is literature saying and what is not yet known? It is important to bring out this point because it highlights the need for your study.

In presenting the educational context of the country, I suggest to link it to your topic. For example, are there policies for girls’ education at the level of post primary education? How does the government promote girls’ education at this level?

The study design is clear and strong. However, I suggest you make it clear that your questionnaire is designed based on literature (if that is so), and you can provide a few references accordingly. The questionnaire can even be attached to the appendix if there are no constraints in relation to wordcount.

Reviewer #3: This is an interesting paper which offers some important insights into the day-to-day barriers to education in rural Burkina Faso. I am reviewing the paper for the first time but I can see that the authors have made good efforts to respond to earlier review comments and have provided a comprehensive overview of how they have responded to these points. I do not feel that any of the findings are surprising or particularly new - but nonetheless they add to a body of literature that captures the complexity of reasons why young people may discontinue attending school in rural communities.

A number of relatively small points I think need addressing:

- The use of 'dropout'; 'dropouts' as labels is derogatory and in my opinion should be avoided.

- There is some conflation/ confusion throughout the paper between what is presented as a 'qualitative study' and issues of sampling and representation. At one stage participants are described as being ' purposefully randomly selected' - ; purposive and random sampling are entirely different processes so there needs to be some greater clarity here.

- While the main the research method was semi-structured interviews - you mention a 'group adapted questionnaire (p.6) - which is described as distinct from the demographic questionnaire. It was unclear to me what this was and what it added to the data

- In terms of ethics and consent - while there is talk of asking adults for their consent and for the consent for participation for young people below the age of 18, there is no mention of young people giving consent to participate in the study or any indication of whether or not they were given any choice about their participation

- The representation of participants via quotes is unclear - so just having the gender and age of a participant does not adequately distinguish between them nor does it tell the reader whether we are hearing from the same young person or whether they are different voices.

7. PLOS authors have the option to publish the peer review history of their article (what does this mean?). If published, this will include your full peer review and any attached files.

Reviewer #1: No

Reviewer #2: No

Reviewer #3: No

---

## [Author Response · Author response to Decision Letter 1]

15 Sep 2022

Reviewer comments

Reviewer #1: Thank you all for addressing my comments. I think this is a nice improvement on the previous version of the paper and you should be commended for your hard work.

We thank the Reviewer for these comments.

Reviewer #2: Girls were seen as lost investments” – stakeholder perspectives on barriers to secondary education in rural Burkina Faso. 

The Reviewer writes “Girls were seen as lost investments…”, which refers to a previous quote in the title of the paper. The quote in the title has been replaced since then, however, to address comments included in the first decision letter we received from the journal (June 2022). This suggests that the Reviewer may, unfortunately, have received an older version of the paper as opposed to our first revision. As a result, not all comments from Reviewer #2 may be entirely applicable. Nevertheless, the comments from Reviewer #2 are still very helpful and we have now incorporated these in a second “harmonized” revision of the paper.

“It’s the poverty” - stakeholder perspectives on barriers to secondary education in rural Burkina Faso” (p.1, title, revised manuscript)

Thank you for giving me the opportunity to review this article. It is an interesting article and relevant for not only the context of Burkina Faso but also for similar countries. The paper is conceptually sound and well-written, but I think it needs some minor revisions before it can be published. The following are my suggestions.

We thank the Reviewer for these comments. 

“These large existing gaps and inequalities in secondary school completion are likely to be further exacerbated by the COVID-19 pandemic, which has resulted in dramatic losses in the income and livelihoods of poor households [19] and temporary school closures in many settings [20].”. This point is relevant, but COVID 19 impact seems to be relatively less a major issue in most African countries, including Burkina Faso. I would suggest also adding and discussing an aspect which is more relevant to education, that is insecurity in many areas of the country. Currently, security issues seem to be more relevant to education than COVID 19 is. How does that affect girls’ education?

Thank you for this suggestion. At the time of data collection in April 2018, the security situation in the study area was just beginning to worsen. Since then, a coup d'état was launched in Burkina Faso in January 2022 and violence around our study area has picked up substantially in the past few years (Nouna area, Boucle du Mouhoun region). Nevertheless, we agree with the Reviewer that this aspect has become more relevant to education (Bene 2022). We have therefore further discussed the security situation in the country, e.g.:

“These large existing gaps and inequalities in secondary school completion are likely to be further exacerbated by the COVID-19 pandemic and worsening security situation, which have resulted in losses of income and temporary school closures in many settings, including in Burkina Faso [20, 21].” (p. 3, revised manuscript) 

“Sixth, since the time of data collection, a coup d'état was launched in Burkina Faso (January 2022) and the security situation has worsened in the study area. Our results, particularly around security as a potential barrier to schooling, may not be generalizable to different periods.” (p. 25, revised manuscript)

References:

Bene K. Gauging secondary school students' terrorism‐related resilience in the Sahel region of Burkina Faso: a quantitative study. Psychology in the Schools. 2022. doi: 10.1002/pits.22779.

After reading the introduction, I could not find out the rationale for the study. In relation to the topic, what is literature saying and what is not yet known? It is important to bring out this point because it highlights the need for your study.

We have now further clarified the rationale in the Introduction section of the paper, as recommended by the Reviewer. First, despite a few recent studies from Burkina Faso, relatively little remains known about the most effective strategies to remove barriers to secondary schooling, particularly in settings where schooling is very low (Psaki S et al., 2022). Second, most prior studies from lower-income countries exclude out-of-school adolescents (such as the Global School-based Student Health Survey). We include the perceptions of a wide range of stakeholders, including out-of-school adolescents.

“Relatively little remains known, however, regarding barriers to secondary schooling, particularly in contexts where schooling is very low. In a recent review of interventions to remove barriers to girls’ schooling in low- and middle-income countries, for example, only two out of >80 studies were conducted in Burkina Faso [22]. Both studies assessed broad efforts to improve the school environment, such as constructing entire schools and additional classrooms, and were focused on the primary school level [23, 24].” (p.3, revised manuscript)

“In contrast to prior studies from low- and middle-income countries [25, 26], we included adolescents and youth who are out-of-school and their parents. In doing so, we included the perceptions of some of the most disadvantaged households [27].” (p.4, revised manuscript)

References:

Psaki S, Haberland N, Mensch B, Woyczynski L, Chuang E. Policies and interventions to remove gender‐related barriers to girls' school participation and learning in low‐ and middle‐income countries: A systematic review of the evidence. Campbell Systematic Reviews. 2022;18(1). doi: 10.1002/cl2.1207.

In presenting the educational context of the country, I suggest to link it to your topic. For example, are there policies for girls’ education at the level of post primary education? How does the government promote girls’ education at this level?

Thank you for this suggestion. We have now provided additional educational context of the region, as recommended by the Reviewer, including in the Introduction and Methods sections. Specifically, we have further highlighted existing evidence on barriers to schooling (UNICEF 2017) and on existing strategies to remove barriers to schooling (Psaki S et al., 2022).

“Relatively little remains known, however, regarding barriers to secondary schooling, particularly in contexts where schooling is very low. In a recent review of interventions to remove barriers to girls’ schooling in low- and middle-income countries, for example, only two out of >80 studies were conducted in Burkina Faso [22]. Both studies assessed broad efforts to improve the school environment, such as constructing entire schools and additional classrooms, and were focused on the primary school level [23, 24].” (p.3, revised manuscript)

“At the national level, commonly reported reasons for school absenteeism include a lack of financial means and a lack of interest in attending school (school is “not deemed necessary”) [34]. These findings suggest that poverty and the perceived benefits of going to secondary school may play an important role in decision-making around school participation and performance [35-37].” (p. 6, revised manuscript)

References:

Psaki S, Haberland N, Mensch B, Woyczynski L, Chuang E. Policies and interventions to remove gender‐related barriers to girls' school participation and learning in low‐ and middle‐income countries: A systematic review of the evidence. Campbell Systematic Reviews. 2022;18(1). doi: 10.1002/cl2.1207.

UNICEF. Rapport d’état du système éducatif national du Burkina Faso, Pour une politique nouvelle dans le cadre de la réforme du continuum d’éducation de base. Pôle de Dakar de IIPE: UNESCO; 2017 [available at: https://unesdoc.unesco.org/ark:/48223/pf0000253643] 

The study design is clear and strong. However, I suggest you make it clear that your questionnaire is designed based on literature (if that is so), and you can provide a few references accordingly. The questionnaire can even be attached to the appendix if there are no constraints in relation to wordcount.

All questionnaires were developed by the study team. The questionnaires covered hypothesized barriers to secondary schooling in the region and were based on readings of the published and grey literature as well as prior research involving out-of-school adolescents in the study area (Trop Med Int Health 2020, Glob Health Action 2010). We have now clarified these points in the Methods section of the paper. Additionally, as recommended by the Reviewer, we have now added all questionnaires to the Appendix (S1 File).

“We developed questionnaires for our qualitative semi-structured interviews separately for each category of stakeholders (e.g., enrolled students, out-of-school adolescents, parents, and teachers). The questionnaires covered hypothesized barriers to secondary schooling, using a priori developed categories of barriers (described below), and were based on readings of the literature and prior research in the study area [28-30]. All questionnaires are available in S1 File. (p. 6, revised manuscript)

“S1 File. Study questionnaires”

References:

De Neve JW, Karlsson O, Canavan CR, Chukwu A, Adu-Afarwuah S, Bukenya J, Darling AM, Harling G, Moshabela M, Killewo J, Fink G, Fawzi WW, Berhane Y. Are out-of-school adolescents at higher risk of adverse health outcomes? Evidence from 9 diverse settings in sub-Saharan Africa. Trop Med Int Health. 2020 Jan;25(1):70-80.

Sié A, Louis V, Gbangou A, Müller O, Niamba L, Stieglbauer G, et al. The Health and Demographic Surveillance System (HDSS) in Nouna, Burkina Faso, 1993-2007. Glob Health Action. 2010;3. doi: 10.3402/gha.v3i0.5284.

Reviewer #3: This is an interesting paper which offers some important insights into the day-to-day barriers to education in rural Burkina Faso. I am reviewing the paper for the first time but I can see that the authors have made good efforts to respond to earlier review comments and have provided a comprehensive overview of how they have responded to these points. I do not feel that any of the findings are surprising or particularly new - but nonetheless they add to a body of literature that captures the complexity of reasons why young people may discontinue attending school in rural communities.

We thank the Reviewer for these comments.

A number of relatively small points I think need addressing: The use of 'dropout'; 'dropouts' as labels is derogatory and in my opinion should be avoided.

Thank you for pointing this out. We have now removed the term throughout the entire paper and have instead used more neutral terms (such as “out-of-school” adolescents). There were two exceptions in the text, however, where we have kept the term drop-out. Specifically, because the term was used as a search term in our literature search (p. 8) and because a respondent’s quote translated to “As I got pregnant, I said I would drop out this year.” (p. 15).

“We included students, out-of-school adolescents, parents of enrolled and out-of-school adolescents.” (p.4, revised manuscript)

“From each selected school, a total of eight interviewees were selected: two enrolled students, two out-of-school adolescents (who did not attend school between one month and one year prior to the interview), one parent of an enrolled student, one parent of an out-of-school adolescent, as well as two teachers.” (p. 6, revised manuscript) 

“Eight out of nine out-of-school adolescents reported being “currently employed” at the time of the study.” (p. 11, revised manuscript)

There is some conflation/confusion throughout the paper between what is presented as a 'qualitative study' and issues of sampling and representation. At one stage participants are described as being ' purposefully randomly selected'; purposive and random sampling are entirely different processes so there needs to be some greater clarity here.

Thank you for bringing this to our attention. We agree with the Reviewer and have now clarified the corresponding sentence, where we write:

“First, we randomly selected five out of 28 secondary schools in the Nouna HDSS.” (p. 26, revised manuscript)

While the main the research method was semi-structured interviews - you mention a 'group adapted questionnaire (p.6) - which is described as distinct from the demographic questionnaire. It was unclear to me what this was and what it added to the data

We apologize for the confusion. We developed questionnaires for our qualitative semi-structured interviews separately for each category of stakeholders (e.g., enrolled students, out-of-school adolescents, their parents, and teachers). These were our study instruments in this qualitative study. An overview of results from the qualitative interviews are presented in Table 2 of the paper. The questionnaires have also been added to the Appendix (S1 File).

Additionally, to assess the background characteristics of respondents who participated in our qualitative interviews (above), we collected basic quantitative socio-demographic characteristics such as age, sex, parental education, primary occupation of the household, and language. An overview of these socio-demographic characteristics is shown in Table 1 of the paper. We have now further clarified these points, where we write – e.g.:

“We developed questionnaires for our qualitative semi-structured interviews separately for each category of stakeholders (e.g., enrolled students, out-of-school adolescents, parents, and teachers). The questionnaires covered hypothesized barriers to secondary schooling, using a priori developed categories of barriers (described below), and were based on readings of the literature and prior research in the study area [28-30]. All questionnaires are available in S1 File. In addition, we collected basic quantitative socio-demographic characteristics from all participants (such as age and parental education).” (p. 6, revised manuscript)

“S1 File. Study questionnaires”

In terms of ethics and consent - while there is talk of asking adults for their consent and for the consent for participation for young people below the age of 18, there is no mention of young people giving consent to participate in the study or any indication of whether or not they were given any choice about their participation.

Written informed consent was obtained from every interviewee or parent for interviewees younger than 18 years. For minors, we sought adult consent in advance of the child’s assent. We defined assent as involving the child to the extent compatible to his or her maturity and with local cultural norms (Cheah & Parker 2014). Although no child in our study expressed reservations about participating in the study, this does not imply that the child necessarily had the final say in the decision to enroll in the study. Enrolment was based on the best interests of the child (World Health Organization 2018). The study was pre-registered and approved by the Institutional Ethics Committee of the Nouna Health Research Center (CRSN), in Burkina Faso, and the Heidelberg University Hospital Ethics Committee, in Germany.

“Written informed consent was obtained from every interviewee or parent for interviewees younger than 18 years. Adult consent was obtained in advance of the child’s assent for minors”. (p. 7, revised manuscript)

References: 

Cheah PY, Parker M. Consent and assent in paediatric research in low-income settings. BMC Medical Ethics. 2014;15(1). doi: 10.1186/1472-6939-15-22.

World Health Organization. Guidance on ethical considerations in planning and reviewing research studies on sexual and reproductive health in adolescents. 2018. [Available at; https://apps.who.int/iris/handle/10665/273792].

The representation of participants via quotes is unclear - so just having the gender and age of a participant does not adequately distinguish between them nor does it tell the reader whether we are hearing from the same young person or whether they are different voices.

Thank you for this suggestion. In our paper, we use illustrative quotes from a wide range of different voices, representing all categories of stakeholders in the study, including in- and out-of-school adolescents, their parents, teachers, school headmasters, and other respondents. Following each quote, we then provide the sex, age, and category of each respondent. For instance, “Out-of-school, female, ages 17 years”. This approach allows the reader to differentiate between the different categories of stakeholders and key demographic characteristics. We also present supplementary quotes in the Appendix (S1 Text). 

Nevertheless, to address the Reviewer’s comment, we have now added the first letter of a pseudonym after each quote to further differentiate between the different voices within each category of stakeholders (e.g., “I” for Isabelle). These pseudonyms allow the reader to track the comments of participants across quotes. For instance, to know whether we are hearing from the same – or a different – out-of-school young woman ages 17 years. We decided against using full names as pseudonyms since full names can have specific social class and ethnic connotations (Wiles 2013). We have now clarified this in the Methods section and added the first letter of pseudonyms after each quote in the Results section - e.g.:

“When showing quotes in the text, we added the first letter of a pseudonym after each quote to protect the privacy of interviewees and allow the reader to track the comments of interviewees across quotes.” (p. 7, revised manuscript)

“It’s the poverty.” – I, out-of-school, female, 17 years

References:

Wiles, Rose. “Thinking ethically: approaches to research ethics.” What are Qualitative Research Ethics? London: Bloomsbury Academic, 2013. 9–24.

---

## [Decision Letter · Decision Letter 2]

4 Nov 2022

“It’s the poverty” – stakeholder perspectives on barriers to secondary education in rural Burkina Faso

PONE-D-21-34581R2

Dear Dr. De Neve,

We’re pleased to inform you that your manuscript has been judged scientifically suitable for publication and will be formally accepted for publication once it meets all outstanding technical requirements.

Kind regards,

Miquel Vall-llosera Camps

Senior Editor

PLOS ONE

Reviewers' comments:

Reviewer's Responses to Questions

**Comments to the Author**

1. If the authors have adequately addressed your comments raised in a previous round of review and you feel that this manuscript is now acceptable for publication, you may indicate that here to bypass the “Comments to the Author” section, enter your conflict of interest statement in the “Confidential to Editor” section, and submit your "Accept" recommendation.

Reviewer #1: All comments have been addressed

Reviewer #2: All comments have been addressed

Reviewer #3: All comments have been addressed

2. Is the manuscript technically sound, and do the data support the conclusions?

Reviewer #1: No

Reviewer #2: Yes

Reviewer #3: Yes

3. Has the statistical analysis been performed appropriately and rigorously? 

Reviewer #1: N/A

Reviewer #2: Yes

Reviewer #3: N/A

4. Have the authors made all data underlying the findings in their manuscript fully available?

Reviewer #1: Yes

Reviewer #2: Yes

Reviewer #3: Yes

5. Is the manuscript presented in an intelligible fashion and written in standard English?

Reviewer #1: Yes

Reviewer #2: Yes

Reviewer #3: Yes

6. Review Comments to the Author

Reviewer #1: Thank you for addressing my comments as well as the comments of the other reviewers. This is a significant improvement on the initial draft and, again, the authors should be commended for their hard work.

Reviewer #2: The authors have addressed all my comments and suggestions. I think paper is worth being published.

Reviewer #3: (No Response)

7. PLOS authors have the option to publish the peer review history of their article (what does this mean?). If published, this will include your full peer review and any attached files.

Reviewer #1: No

Reviewer #2: **Yes: **Jean-Baptiste M.B. SANFO

Reviewer #3: **Yes: **Elaine Chase

---

## [Editor Report · Acceptance letter]

9 Nov 2022

PONE-D-21-34581R2 

“It’s the poverty” – stakeholder perspectives on barriers to secondary education in rural Burkina Faso 

Dear Dr. De Neve:

I'm pleased to inform you that your manuscript has been deemed suitable for publication in PLOS ONE. Congratulations! Your manuscript is now with our production department. 

Kind regards, 

on behalf of

Dr. Miquel Vall-llosera Camps 

Staff Editor

PLOS ONE